# The impact of computer science education in primary schools: Evidence from a randomized controlled trial in Iraq

Satoshi Shimizutani[1], Shimpei Taguchi[2]*, Hiroyuki Yamada[3]

1 Graduate School of International Development, Nagoya University, Nagoya, Aichi, Japan, 2 Faculty of Political Science and Economics, Takushoku University, Bunkyō, Tokyo, Japan, 3 Faculty of Economics, Keio University, Minato, Tokyo, Japan

☯ These authors contributed equally to this work.
*staguchi@ner.takushoku-u.ac.jp

## Abstract

With the growing digitization of society, there is a need to enhance computational thinking as an indispensable skill for modern daily life. Consequently, computer science education for children at early ages has become increasingly important. This study conducts a randomized controlled trial to examine the impact of the interventions using educational robotics as well as computer-aided mathematics drills (via a "math app") on students' performance in primary schools in Basra, Iraq. We provide several new empirical findings. First, the short-run impact of robotics-based learning on computational thinking is positive and statistically significant for girls, particularly poor performing girls, but not for boys. Second, the impact on computational thinking is augmented by introducing a math app, further improving computational thinking. Together, these two interventions also enhance general intelligence. Third, the positive impact was still evident more than three months after the interventions for girls who received both computer science and math education, suggesting their complementarity. Our results show that computer science education using educational robots in primary schools is effective in enhancing computational thinking and relevant skills.

## Section 1: Introduction

Over the past two decades, there has been a growing recognition that computational thinking (CT) is an essential skill for modern society. Computational thinking is broadly considered to be the underlying problem-solving cognitive process that enables new read-write practice with computer programming languages and is referred to as "code literacy" [1]. Code literacy is not required exclusively for computer scientists' activities and can be widely applied in daily life [2,3]. In a seminal paper that revived computational thinking this century, Wing [4] argued that

**Data availability statement:** The data pertaining to this study are owned by JICA Ogata Research Institute. The data for this part of the study can be requested by calling the JICA Ogata Research Institute at +81-3-3269-2911 or through the Corresponding Author.

**Funding:** This work was supported by JICA Ogata Sadako Research Institute for Peace and Development (no grant number available). The funder had no role in the study design, data collection and analysis, decision to publish, or preparation of the manuscript.

**Competing interests:** The authors have declared that no competing interests exist.

computational thinking is a universally applicable skill set for everyone to absorb and make use of, while Grover and Pea [5] pointed out some confusion over the definition of computational thinking in related papers, which is still the case to date [3]. The idea of computational thinking goes back to the 1960s, as Alan Perlis emphasized the need to learn programming for all college students [6]. In the 1980s, Seymour Papert pioneered the idea of children developing procedural thinking [7], and now its importance and usefulness is widely acknowledged in a world that has experienced rapid digitalization driven by pervasive and foundational processes across the whole breadth of society [8].

Under these circumstances, the development of computational thinking requires a growing focus in school curriculums as educational innovation and computer science education in children of early ages gain importance [9,10]. Many countries introduced computational science education to foster computational thinking in their national educational curriculum as a set of problem-solving skills to be acquired by school students [1,11]. Moreover, some countries have expanded computer science education in primary school curriculums, such as England (2014), Finland (2016) and Japan (2020), to develop computational thinking from earlier ages.

There have been a growing number of research papers on computational thinking. Moreno-León et al. [11] reveal that the introduction of computer programming with Scratch, which provided coding activities, was effective in improving academic performance for 6th-grade students but not 2nd-grade students and the effect differs when it is integrated with social studies or mathematics. Angeli and Valanides [12] show that the effect of educational robotics is augmented by scaffolding techniques that benefit boys (card activity) and girls (collaborative writing activity) differently. Tengler et al. [13] conclude that combining stories, texts, and literature with educational robotics offers a promising approach for students to absorb computational skills. In addition, there are a limited number of studies with very small sample sizes (e.g., typically less than 100 targets) that identify the positive correlation between programming-related intervention and mathematics-related outcomes at the primary school level [14–16] and middle school level [17,18].

However, studies that provide evidence on the impact of computer science education on primary school students are quite limited, with small sample sizes, despite the great interest of both academics and policymakers. We suggest several reasons for the lack of evidence. The introduction of computer science education in young students in public schools is still novel, and therefore, there are no universally adopted methodologies or standardized materials to provide computer science education in the primary school curriculum. Moreover, there are no validated tests to measure the impact of computer science education on computational thinking exclusively for primary school children.

This study quantitatively examines the impact of computer science education on the computational thinking of students at early ages. We focus on a robotics-based learning project for computer science education in primary schools in Basra, the second-largest city in Iraq. The use of educational robotics has been an increasing trend in recent years because they are inexpensive, equipped with multi-functions,

and well-received by students [13]. We conduct a randomized controlled trial (RCT) to measure the impact of computer science education for grade-4 students on computational thinking using a newly developed validated test to uncover the effectiveness of the project. We aim to contribute to the literature in four ways.

First, rigorous evidence on the impact of computer science education on primary school students is very scarce, in contrast to the available literature on junior-high children. This likely reflects the lack of availability of valid tests to measure computational thinking skills for younger students, despite the recognition that earlier intervention in computer science education is considered more effective for developing cognitive skills. We utilize a newly developed valid test exclusively for younger children (aged 8–10) and well suited to robotics-based learning to measure computational thinking. To our knowledge, there have been no previous large-scale quantitative studies on the impact of computer science education on computational thinking empowered by a randomized controlled trial (RCT) in primary schools.

Second, in addition to a simple impact evaluation of robotics-based learning, we investigate the possible synergistic impact of implementing multiple interventions in computer science and mathematics education using a math app. The main focus of our computer science education project using robots is to acquire skills in "sequences" and "conditionals," abilities that are potentially augmented by logical thinking boosted by mathematics education. Thus, we set up two interventions and divided the sample into four groups: (1) both computer science education and math education, (2) computer science education only, (3) math education only and (4) neither of those two interventions (control groups). Through these interventions, we aim to explore the complementarity of robotics-based computer science learning and math education.

Third, we measure the impact of the interventions not only immediately after the treatments were conducted but also three or more months later to explore the sustainability and kinds of longer-term impact. By estimating the impact at two different intervals, we confirm whether a student absorbed computational thinking in a sustainable way. To our knowledge, there has been little research to measure both immediate and longer impacts of computer science education at different times in which the sample is confined to primary school students.

Fourth, empirical evidence on any projects in Iraq has been very scarce, a commonality not limited to the field of education. Since many women live under conservative social norms related to Muslim traditions in the country, any gender disaggregated evidence is useful for understanding the direction of the education policy. Moreover, we believe this study is one of the few attempts to conduct a large-scale randomized controlled trial for impact evaluation in Iraq.

This paper proceeds as follows. The next section explains the materials used in this study, including the interventions and measurement tests. Section 3 describes our research design and data collection. Section 4 presents the empirical strategy, and Section 5 reports and discusses the empirical results. The last section summarizes the key findings and offers recommendations for future research.

## Section 2: Materials for our research

### Robotics-based learning and math apps

This study comprises two interventions. The main target intervention is robotics-based learning of computer science education to promote computational thinking in the classroom. Robotics-based learning offers a feasible approach to computer science education at the primary level [5,13]. Several studies have shown that educational robotics is suitable for encouraging children to program while also promoting problem-solving and computational thinking with no differences in age or gender [19–21]. Robotics-based learning provides a valuable opportunity to equip learners for coding, the core skill of computational thinking, and is considered particularly effective for young learners [22].

Among currently available educational robotics, we use the Ozobot in this study. An Ozobot is an educational coding robot that moves on two wheels to follow lines. An Ozobot is equipped with colour sensors that recognize colour codes on lines, while colour programming controls the speed, direction and stopping point of an Ozobot with different colour dots. Among the different levels of programming of Ozobot, this study uses a paper-based version suitable for primary school students to learn computer programming due to its ease of handling and appearance [23,24]. The paper version of Ozobot

programming can be implemented easily with pens and papers only and cultivates skills relevant to the use of sequences and conditionals among computational concepts [13,25].

The other intervention of this study is mathematics education with computer-aided drills. Hereafter, we call these drills "math app." The math app used in this study is a paper-based version of a math drill developed by JICA (Japan International Cooperation Agency). The drill was designed to teach "numbers and calculations" in mathematics for grade 4 students to cover four arithmetic operations of integers of up to three digits (mostly one or two digits). The math app was installed into a tablet that each student in the sample used repeatedly in the classroom.

## Test materials

We utilize three tests in this study. The same tests are used at both baseline and endline to measure the difference in students' performance before and after the intervention.

The first test is the Beginners Computational Thinking Test (BCTt for short) [3]. This test was developed specifically for primary school students. While there have been many attempts to assess computational thinking to introduce computational thinking in the school curriculum, most focus primarily on middle school grades and above [26] or on specific programming environments [3]. The BCTt was developed for five to twelve-year-old students from the Computational Thinking Test (CTt for short) by Román et al [1]. It is a self-contained instrument independent of any programming environment. The BCTt adopted Brennan's 3D framework basic computational concepts [25] and adapted the CTt designed for 10–16-year-old students—in both form and content—for young students. The main task is to carry a chick along to its mother (the hen) through a maze, in either small or large format and in the correct order. Students need to avoid obstacles, such as a cat, and collect items along the way, such as another chicken. There are 25 questions in the BCTt. Note that both CTt and BCTt have been repeatedly validated by experts in many settings [3].

The second test is the math test. The math test was created from the check tests developed together with the math drill by JICA. The purpose of the check test was to evaluate whether a student can perform four basic arithmetic operations successfully after learning the math drill. The math test comprised 30 questions ordered in terms of difficulty.

The third test, originally developed by John Raven in the 1930s, is Raven's Progressive Matrices (RPM), which measures general human intelligence and abstract reasoning. RPM is regarded as suitable to estimate fluid intelligence [27]. We included this test because computer science and math education may enhance not only computational thinking and math ability but also improve human intelligence in general. The RPM is a non-verbal, multiple-choice test using images and patterns. It has been widely used as a psychometric instrument to quantify ability to solve problems and think logically, regardless of an individual's skill of language or culture. The questions in RPM consist of geometric pictures with one missing piece, and each test-taker is asked to identify the missing piece from six to eight choices to complete a picture.

Among the several versions of RPM, we use Standard Progressive Matrices (RSPM), which includes the shortest number of questions and is considered suitable for our intervention. This version contains twelve questions, and the questions are listed in order of increasing complexity and difficulty [28].

## Section 3: Research design and data collection

### Primary schools in Iraq

The national education system was initiated in Iraq in 1921. All public schools became free in the early 1970s, making schooling available to all children. Although Iraq's educational level was widely recognized as the highest in the Middle East until the 1980s, it is said to be declining due to the inefficiency of the educational system, the outflow of human resources caused by the long-standing conflict, and the deterioration of educational facilities and equipment [29].

Structurally, the education system in Iraq comprises six years of primary education— mandatory under the constitution—followed by three years of lower secondary and three years of upper secondary education. There are 160 school

days annually in Iraq (32 weeks; 5 days). In primary schools, students have 27–28 class periods per week, of which 5–6 are mathematics [30].

According to UNICEF [31], the completion rate for primary school in Iraq is 76% (78% male; 73% female). Of children who do not complete primary school, 53% are female students; 9% of female children and 7% of male children are out of school at the primary school level. These data suggest that female students are more disadvantaged in primary education.

## Research design

The target project of this study was conducted for forty primary schools in the Zubair District of Basra Prefecture. The Zubair Educational Directorate office provided a list of forty-eight primary schools located in Zubair District in summer 2022, with some information on the number of students in school and class shifts. Among these schools, we selected forty primary schools by excluding schools with extremes in terms of the numbers of students and classes or schools with evening classes to ensure homogeneity between schools in the sample.

Then, we randomly allocated those forty primary schools into four groups, each with ten schools. The first group includes schools that received both robotics-based computer science education and mathematics education using the math app. The second group received robotics-based learning only, while the third group had the math app learning only. The fourth group did not offer either computer science or mathematics education. The primary schools in Zubair District are dispersed geographically, and most students in different schools do not usually communicate with each other regularly, at least during our intervention. Thus, we judge that any spillover effect of intervention in schools is negligible and did not contaminate the main results in this study.

At the baseline, 40 students enrolled in grade 4 from each school were randomly selected due to limitations in the availability of equipment used in the intervention (Apple iPad tablets). No school in the sample had less than forty students in grade 4. As a result, the number of total students in our sample is 1,600 at the baseline. However, some students in the sample did not take the endline tests. Thus, the sample size used in the analysis in the following section is smaller than 1,600.

## Data collection

The field research started in 13 November 2022 and ended in 2 May 2023. We obtained the consent of the students' teachers for this study and also the consent of the parents of each surveyed individual through their teachers. These consents were obtained verbally, which procedure was approved by the Institutional Review Board, Institute for Economic Studies, Keio University. First, the enumerators visited each school and administered the baseline survey—i.e., BCTt, math and PRM tests—to students in the sample. As described above, interventions for each sample school started after the baseline survey. The interventions were implemented by three different teams of enumerators, with a maximum of three weeks allowed for each school, including the baseline survey, intervention, and endline survey.

For the schools in the first and second groups (20 schools in total) to benefit from computer science education, the enumerators conducted two-day classes (45 minutes per class) of computer science education using educational robots (Ozobots). On the first day, the enumerators explained the purpose of computer science education and introduced the concept of coding and algorism. Then, they taught the coding pattern with some examples of algorism using "origami", which is the traditional Japanese art of paper folding, and instructed students on how to play a "route game" in which students were required to move a robot from start to end points using programming. On the second day, enumerators explained the programming using an Ozobot in practice, as well as line tracing and colour codes. Finally, the students who benefited from robotics-based learning programmed Ozobots to play the route game, moving from a starting point to a designated goal.

On the other hand, the intervention for students in the first and third group schools involved using the math app. The students in the first group were instructed on computer science education first, then received a math workshop using the math app later. The enumerators visited for seven consecutive school days to perform a session of mathematics

education using the math app. The mathematics workshop comprised seven sections, and each section was conducted for 45 minutes per day.

After the interventions of computer science and/or mathematics education, the students in the sample took the same tests at the endline as in the baseline. As noted above, some students who took tests at the baseline did not take exams at the endline with an attrition rate of 8.4%. In addition, we conducted the second endline tests in April 2023 for the students who received any interventions before the end of 2022 to explore any longer impact from the interventions. These students comprise 60% of all students in the sample—a total of 960 students. The duration between the intervention and the second endline was at least three months, with an attrition rate of 13.1%. We regressed a dependent variable to take 1 for those who dropped out at the endline on the observable characteristics in Table 1 and confirmed that any coefficients are not statistically significant. This is also the case for the second endline.

## Section 4: Empirical strategy

This study employs a randomized controlled trial (RCT) to formally examine the project's impact. To our knowledge, the methodology is the most suitable for a causal inference of the impact of an intervention. At the baseline, the forty schools

**Table 1. Balance test.**

| | Programing+Math | Programing only | Math only | Control | p-value all equal |
|---|---|---|---|---|---|
| | (1) | (2) | (3) | (4) | (5) |
| *Number of observations: 1,465* | *361* | *364* | *374* | *366* | |
| **Outcomes** | | | | | |
| BCT test score (25 points) | 8.66 | 8.60 | 9.07 | 8.07 | 0.11 |
| | (5.67) | (5.59) | (5.73) | (5.17) | |
| Mathematics test score (30 points) | 11.01 | 11.31 | 11.71 | 11.78 | 0.23 |
| | (5.46) | (6.11) | (5.53) | (5.76) | |
| Raven test score (12 points) | 5.97 | 5.56 | 5.64 | 5.60 | 0.35 |
| | (3.24) | (3.50) | (3.49) | (3.38) | |
| **Covariates** | | | | | |
| Female | 0.40 | 0.38 | 0.40 | 0.41 | 0.90 |
| | (0.49) | (0.49) | (0.49) | (0.49) | |
| Age | 10.01 | 10.11 | 10.12 | 10.14 | 0.17 |
| | (0.86) | (0.88) | (0.83) | (0.82) | |
| Ever repeated | 0.21 | 0.18 | 0.17 | 0.16 | 0.40 |
| | (0.41) | (0.39) | (0.37) | (0.37) | |
| Engaged in commercial activities when not at school | 0.30 | 0.37 | 0.36 | 0.33 | 0.26 |
| | (0.46) | (0.48) | (0.48) | (0.47) | |
| Taking care of younger children when not at school | 0.71 | 0.74 | 0.70 | 0.69 | 0.53 |
| | (0.45) | (0.44) | (0.46) | (0.46) | |
| Internet available at home | 0.82 | 0.86 | 0.84 | 0.87 | 0.22 |
| | (0.39) | (0.35) | (0.37) | (0.34) | |
| Father working | 0.88 | 0.91 | 0.91 | 0.89 | 0.30 |
| | (0.32) | (0.28) | (0.28) | (0.32) | |
| Mother working | 0.17 | 0.20 | 0.24 | 0.22 | 0.18 |
| | (0.38) | (0.40) | (0.43) | (0.41) | |

***p<0.01,

**p<0.05,

*p<0.1. All covariates except age are dummy variables.

in our sample were randomly assigned to one of the four groups (ten schools each). The key assumption of an RCT is that each assigned group is homogenous on average and they do not differ from each other before the intervention.

Thus, we perform a balance test among four groups at the baseline. Table 1 reports the results. The first three rows present the outcome variables that comprise the three test scores: the BCTt score for computational thinking (maximum of 25 points), the mathematic test score (30 points) and the Raven test score for general intelligence (12 points). We perform a statistical test to the null hypothesis that the average of each test score in different groups is equal to each other. We cannot reject the joint null hypothesis that the three test scores are equal. In other words, four different groups are not heterogeneous in terms of outcome variables. Moreover, we performed an analysis to detect any statistical difference in the observable characteristics of students in each group, including gender, age, repetition, outside school activities and the job status of parents. The remaining rows of Table 1 show that no significant difference among groups is detected in observable characteristics of students. Thus, we judge that we were successful in the random allocation into four groups with a homogenous background.

Next, we turn to argue our empirical strategy. We start with the basic specification to obtain the average treatment effect (ATE) of the project. Since our research design is a randomized controlled trial (RCT), the impact is in principle estimated by "difference-in-mean" at the end-line by comparing the average effect of the intervention in the treatment group with that in the control group. Note that the compliance rate in a treatment group is not 100% in most cases of RCTs, and thus, the impact is captured by the "intention-to-treat" (ITT) effect because not all individuals in the treatment group have genuinely participated in the program. In contrast to previous research, all students in each group in this study indeed benefited from the assigned intervention (computer science education and/or math apps). Thus, we estimate the average treatment effect to measure the impact of the intervention precisely, unaffected by selection bias into being treated. This characteristic is one of the advantages of this study in asserting that the impact is captured without considering non-compliance.

Our basic specification is written as follows:

$$Y_{ij} = \alpha + \beta_1 Both_j + \beta_2 Programming_j + \beta_3 Math_j + \epsilon_{ij} \tag{1}$$

where; $Y_{ij}$ stands for the outcome variable (each standardized score for three different tests) for student $i$ in school $j$ at the end-line. $\alpha$ is a constant term. $Both_j$ is a dummy indicating whether a student attended a school that received both computer science education and math education or not; if a student in a school for both interventions, $Both_j$ takes 1 and otherwise 0. Similarly, $Programming_j$ is a dummy indicating whether a student attended a school that received the computer science education only but not the math education and $Math_j$ is an indicator of whether a student attended a school that received the math education only, not the computer science education. $\epsilon_{ij}$ is a well-behaved error term. The coefficients of interest are $\beta s$ which measures the impact of each intervention.

We develop this simple basic specification to incorporate observable characteristics of students as well as dummies of the enumerator teams to pursue a granular effect. The observable characteristics of students are shown in Table 1. Moreover, the enumerators were divided into three teams to visit schools to conduct the interventions. By including dummy variables for each enumerator team, we can control for any differences between the enumerator teams in terms of teaching or grading that are not relevant to student characteristics because Villalustre and Cueli [32] showed that characteristics of instructors may affect students' performance.

Therefore, our specification strategy is described as follows.

$$Y_{ij} = \alpha + \beta_1 Both_j + \beta_2 Programming_j + \beta_3 Math_j + \gamma_1 Y_{ijt-1} + \gamma_2 X_{ijt-1} + \delta E_j + \epsilon_{ij} \tag{2}$$

where the notations are the same as in specification (1) except that the dependent variable at baseline (the lagged dependent variable $Y_{ijt-1}$ at the baseline) and the characteristics of students $X_{ijt-1}$ as well as team of dummy variable for each team of enumerators $E_j$ are now included as covariates.

In what follows, we report the estimation results using specification (2), which we prefer to (1) for brevity. We confirmed that the results are intact if we employ specification (1). In the regression analysis, we limit our sample to the students who participated in both baseline and the end-line surveys. All standard errors that are reported are clustered at the school level.

## Section 5: Empirical results and discussion

Table 2 reports the regression results on the impact of the intervention at the first endline. This impact captures the immediate effect right after implementing the computer science and/or math education. Each column presents the coefficients of interest to measure the impact of the intervention, which corresponds to the average treatment effect in each test score.

The first row represents the coefficients on the interaction term of computer programming and math education, which stands for the impact on the first group that received both robotics-based and math app-based learning. The second row reports the coefficients on the dummy variable for the second group that accepted the computer science program only while the third one is the coefficients for the third group that received the math education only.

The first three columns show the impact of the intervention on the scores for the BCTt, math and Raven tests. The coefficient is positive and significant in the computational thinking test for students who received both interventions but not so for groups who received only one treatment (Column (1)). The same pattern is observed in the math test (Column (2)); the coefficient is positive and significant for students who received both programs. In the Raven test, the coefficient is positive and statistically significant for students who took computer science education only. These results indicate that the combination of computer science and math education improves test scores for both computational thinking and mathematics.

The sample of regression results in Columns (1) to (3) is girls and boys combined. When we disaggregate the sample by gender, a different and contrasting picture emerges. For girls (Columns (4) to (6)), the coefficient is positive and significant in all tests for those who received both computer science and math education. The coefficient is positive in all tests but only significant in the Raven test for girls who learned only computer science education. The size of the coefficient in the BCTs is large but imprecisely estimated. The coefficient is positive in all tests and significant in the math and Raven tests for girls who learned math education aided by the math app (and no computer science education).

**Table 2. Impact of the interventions at the endline.**

|  | All | | | Girls | | | Boys | | |
|---|---|---|---|---|---|---|---|---|---|
|  | BCT | Math | Raven | BCT | Math | Raven | BCT | Math | Raven |
|  | (1) | (2) | (3) | (4) | (5) | (6) | (7) | (8) | (9) |
| Programing + Math | 0.187* | 0.325** | 0.122 | 0.373* | 0.664*** | 0.494*** | 0.176 | 0.191 | 0.002 |
|  | (0.102) | (0.145) | (0.096) | (0.191) | (0.194) | (0.112) | (0.123) | (0.180) | (0.103) |
| Programing only | 0.051 | 0.065 | 0.157* | 0.501 | 0.409 | 0.438** | −0.060 | −0.045 | 0.083 |
|  | (0.162) | (0.157) | (0.092) | (0.332) | (0.403) | (0.204) | (0.182) | (0.148) | (0.075) |
| Math only | −0.062 | 0.155 | 0.118 | 0.198 | 0.496** | 0.525*** | −0.071 | 0.066 | −0.028 |
|  | (0.119) | (0.165) | (0.105) | (0.180) | (0.230) | (0.165) | (0.137) | (0.199) | (0.089) |
| N. of obs. | 1,465 | 1,465 | 1,465 | 586 | 586 | 586 | 879 | 879 | 879 |
| Adjusted R-squared | 0.521 | 0.555 | 0.417 | 0.529 | 0.555 | 0.405 | 0.522 | 0.585 | 0.441 |

(Note) Robust standard errors clustered at the school level are reported in parenthesis.

***$p < 0.01$,

**$p < 0.05$,

*$p < 0.1$. All regressions include the covariates shown in Table 1. The outcome variables are standardized with reference to the baseline scores of the control group.

These results reveal that the intervention had a favourable impact for girls. The computer science education contributed significantly to improving all test scores when combined with the math apps education and enhanced general human intelligence even when not combined with math education. The mathematics education encouraged a significant increase in test scores in the math and Raven tests. In contrast, all coefficients are imprecisely estimated and not statistically significant for boys, though the sample size of boys is larger than girls (Columns (7) to (9)). The coefficient is positive but not significant even for boys who received both programming and mathematics education, in contrast to girls.

We observed that the computer science education has a positive and significant impact for girls when combined with the math education, but this is not the case for boys. In order to scrutinize the impact for each gender, we divide the sample further by the median of the test scores at the baseline because the impact may depend on the students' pre-performance before the intervention. We use the common criteria to group the sample for the three tests comprising the median of the math test at the baseline. We confirm that the estimation results are not much affected if we group the sample by the median of each test at the baseline. Our decision is affirmed in research by Clements et al. (2016), which found that executive function—the cognitive ability to control, monitor or regulate one's thinking and behaviour that is associated with achievement in young children—was consistently associated with children's mathematical skills.

Table 3 reports the estimation results dividing the sample at the median. Columns (1) to (3) show the coefficients for girls whose test scores were above the median and (4) to (6) for those below the median. For girls above median, we detect a clear difference among girls above and below the median. The coefficients are positive but not significant in the BCTt for any groups who received any treatments, including those that benefited from both programming and math education. The coefficient is positive and significant in the math test for the groups who received both education and math education only. The coefficient in the Raven test is positive and significant when students took the math education only and not significant for girls who received computer science education.

In contrast, the positive impact is clearly observed for girls whose performance was below the median at the baseline. Girls who received both computer science and math education overperformed in all three tests. The intervention improved both BCTs and the Raven scores for girls who received the robotics-based programming only. Moreover, the treatment increased both the math and Raven scores for girls who received the math apps education only. In other words, a single intervention (either computer science or math education) contributed to improving their own target test and the Raven test.

**Table 3. Impact of the interventions at the endline by median.**

| | Girls above median | | | Girls below median | | | Boys above median | | | Boys below median | | |
|---|---|---|---|---|---|---|---|---|---|---|---|---|
| | BCT | Math | Raven | BCT | Math | Raven | BCT | Math | Raven | BCT | Math | Raven |
| | (1) | (2) | (3) | (4) | (5) | (6) | (7) | (8) | (9) | (10) | (11) | (12) |
| Programing + Math | 0.107 | 0.483** | 0.146 | 0.580** | 0.840*** | 0.838*** | 0.241** | 0.341** | 0.031 | 0.137 | −0.042 | 0.016 |
| | (0.145) | (0.218) | (0.115) | (0.208) | (0.187) | (0.140) | (0.102) | (0.155) | (0.091) | (0.200) | (0.214) | (0.130) |
| Programing only | 0.209 | 0.378 | 0.069 | 0.822** | 0.382 | 0.824*** | 0.070 | −0.009 | 0.086 | −0.179 | −0.220 | 0.085 |
| | (0.254) | (0.509) | (0.252) | (0.374) | (0.333) | (0.215) | (0.146) | (0.141) | (0.082) | (0.230) | (0.193) | (0.133) |
| Math only | 0.140 | 0.508* | 0.317*** | 0.314 | 0.518** | 0.764*** | −0.027 | 0.144 | 0.004 | −0.157 | −0.121 | −0.083 |
| | (0.138) | (0.243) | (0.106) | (0.207) | (0.224) | (0.198) | (0.136) | (0.156) | (0.082) | (0.192) | (0.243) | (0.162) |
| N. of obs. | 323 | 323 | 323 | 263 | 263 | 263 | 472 | 472 | 472 | 407 | 407 | 407 |
| Adjusted R-squared | 0.416 | 0.332 | 0.296 | 0.499 | 0.325 | 0.406 | 0.498 | 0.415 | 0.397 | 0.332 | 0.296 | 0.309 |

(Note) Robust standard errors clustered at the school level are reported in parenthesis.

\*\*\*p<0.01,

\*\*p<0.05,

\*p<0.1. All regressions include the covariates shown in Table 1. The outcome variables are standardized with reference to the baseline scores of the control group.

It is noticeable that the size of the impact on general intelligence measured by the Raven test is comparable among those three groups.

On the other hand, Columns (7) to (12) in Table 3 show the coefficients for boys. For boys above the median, we see some positive effects on the BCTt and the math tests for students who received both computer science and math education, but the remaining coefficients are imprecisely estimated and not statistically significant (Columns (7) to (9)). Moreover, all coefficients are not statistically significant for boys whose test performance is below the median at the baseline. The coefficients are generally smaller and not different from zero (Columns (10) to (12)). In contrast to girls, most of the coefficients are not statistically significant for boys except those above the median who received both treatments in the BCTs and the math tests.

So far, we examined the immediate impact right after the intervention. Now, we turn to investigate the longer impact. We conducted the second endline survey for students who received the intervention by the end of 2022. The sample size of those students is 960. The second endline survey using exactly the same three tests was conducted in April 2023, three to five months after the intervention.

Table 4 shows the estimation results. For girls and boys combined, the coefficients are not statistically significant and no systematic pattern is observed (Columns (1) to (3)). When we divide the sample by gender, we see the positive and significant coefficients in all tests for girls who received both computer science and math education (Columns (4) to (6)). The size of the coefficient is smaller in the math and Raven tests than those in Table 2 but comparable in the BCT test. The remaining coefficients are not significant, showing that a single intervention did not have a positive impact over a couple of months except for the significant impact of the math education on the Raven test. For boys, we do not see any significant coefficients (Columns (7) to (9)) regardless of the pre-performance. This lack of significance was expected because there was no immediate impact of the intervention for boys.

Table 5 reports the regression results by dividing the sample at the median from the baseline in line with Table 3. The left-half columns show the coefficients for girls (Columns (1) to (6)) and the right-half for boys (Columns (7) to (12)). For girls, the coefficients are not statistically significant for those whose test scores were above the median at the baseline. While those are positive in the BCT test, they are estimated imprecisely. In contrast, the coefficient for girls whose test scores were below the median is positive and significant for those who received both computer science and mathematics

**Table 4. Impact of the interventions at the second endline.**

|  | All | | | Girls | | | Boys | | |
|---|---|---|---|---|---|---|---|---|---|
|  | BCT | Math | Raven | BCT | Math | Raven | BCT | Math | Raven |
|  | (1) | (2) | (3) | (4) | (5) | (6) | (7) | (8) | (9) |
| Programing + Math | 0.125 | 0.205 | 0.157 | 0.443** | 0.297** | 0.296*** | −0.381 | −0.013 | 0.088 |
|  | (0.223) | (0.121) | (0.101) | (0.156) | (0.136) | (0.080) | (0.272) | (0.122) | (0.106) |
| Programing only | −0.061 | −0.066 | 0.093 | 0.194 | −0.033 | 0.115 | −0.464 | −0.204 | 0.091 |
|  | (0.187) | (0.166) | (0.119) | (0.212) | (0.303) | (0.161) | (0.276) | (0.134) | (0.091) |
| Math only | −0.045 | 0.105 | 0.206 | 0.092 | 0.166 | 0.355* | −0.343 | −0.050 | 0.139 |
|  | (0.176) | (0.144) | (0.151) | (0.125) | (0.186) | (0.187) | (0.289) | (0.132) | (0.102) |
| N. of obs. | 834 | 834 | 834 | 414 | 414 | 414 | 420 | 420 | 420 |
| Adjusted R-squared | 0.346 | 0.542 | 0.301 | 0.407 | 0.547 | 0.306 | 0.318 | 0.537 | 0.272 |

(Note) Robust standard errors clustered at the school level are reported in parenthesis.

***$p < 0.01$,

**$p < 0.05$,

*$p < 0.1$. All regressions include the covariates shown in Table 1. The outcome variables are standardized with reference to the baseline scores of the control group.

**Table 5. Impact of the interventions at the second endline by median.**

| | Girls above median | | | Girls below median | | | Boys above median | | | Boys below median | | |
|---|---|---|---|---|---|---|---|---|---|---|---|---|
| | BCT | Math | Raven | BCT | Math | Raven | BCT | Math | Raven | BCT | Math | Raven |
| | (1) | (2) | (3) | (4) | (5) | (6) | (7) | (8) | (9) | (10) | (11) | (12) |
| Programing + Math | 0.204 | 0.047 | 0.092 | 0.574*** | 0.541*** | 0.484*** | −0.284 | 0.100 | 0.137 | −0.365 | −0.144 | 0.158 |
| | (0.177) | (0.144) | (0.080) | (0.177) | (0.151) | (0.114) | (0.293) | (0.117) | (0.115) | (0.250) | (0.160) | (0.172) |
| Programing only | 0.050 | −0.199 | −0.065 | 0.144 | 0.101 | 0.204 | −0.374 | −0.188 | 0.319*** | −0.292 | −0.213 | 0.042 |
| | (0.221) | (0.245) | (0.185) | (0.110) | (0.277) | (0.127) | (0.321) | (0.119) | (0.099) | (0.303) | (0.165) | (0.182) |
| Math only | 0.023 | 0.197 | 0.281 | 0.256*** | 0.231 | 0.458** | −0.492 | −0.034 | 0.158 | −0.143 | −0.057 | 0.251 |
| | (0.149) | (0.141) | (0.216) | (0.045) | (0.205) | (0.170) | (0.309) | (0.119) | (0.139) | (0.334) | (0.231) | (0.189) |
| N. of obs. | 211 | 211 | 211 | 203 | 203 | 203 | 228 | 228 | 228 | 192 | 192 | 192 |
| Adjusted R-squared | 0.261 | 0.343 | 0.165 | 0.373 | 0.281 | 0.369 | 0.300 | 0.443 | 0.245 | 0.188 | 0.170 | 0.220 |

(Note) Robust standard errors clustered at the school level are reported in parenthesis.

***$p < 0.01$,

**$p < 0.05$,

*$p < 0.1$. All regressions include the covariates shown in Table 1. The outcome variables are standardized with reference to the baseline scores of the control group.

education, showing that the impact of the multiple interventions persists for a couple of months. However, this is not the case for girls who took computer science education but not math education, suggesting that a single intervention is not sufficient to sustain a positive effect and the synergistic effect helped to prolong the impact of computer science education. In addition, the coefficient for girls who took only the math test is positive and significant in the BCTs and Raven test, though positive but not significant in the math test.

In contrast, the coefficients are not statistically significant for any groups in all test scores for boys except for the Raven test for boys above the median who benefited from the computer science education only. No systematic pattern in the coefficients is observed. This is expected given that the positive effect was not observed immediately.

These results reveal several interesting new findings. First, the immediate impact of the robotics-based learning of computer science education is effective for improving computational thinking for girls and the impact is pronounced for girls whose pre-performance was poor. The positive impact of the intervention is found only for girls, not boys. Our result contrasts with Atmatzidou and Demetriadis [19], who found no gender difference in the impact of educational robotics learning activity in secondary schools in Greece. We argue that the favourable impact for girls stems from their unequal access to quality education compared to boys [33]. This finding is consistent with existing research [34,35], suggesting that girls, who have been considered to be worse than boys, are as capable as boys when given meaning full opportunity to learn.

Second, computer science education contributed significantly to improving all test scores when combined with mathematics education using the math app. This finding suggests a synergistic effect, as the impact of computer science education could be augmented by a combination of other alternatives. At the same time, computer science education enhances general intelligence even when not combined with the math education, suggesting that the robotics-based learning of computer science education improves not only computational thinking but also general intelligence. This is also the case for the math education using the math app, which encouraged a significant increase in test scores in the math and Raven tests. Thus, we observe some synergistic effects across interventions apart from a single intervention. Transfer of cognitive skills, including creativity, reasoning, and mathematical skills from programming education has long been discussed. Scherer et al. [36] conducted a meta-analysis identifying positive transfers of such skills from programming education. Our finding is consistent with this previous research.

Third, the positive effect lasts at least three months if the computer science education is combined with the math education. This finding reveals that a multiple intervention is more effective in consolidating new knowledge than a single intervention if computer science education is jointly conducted together with others [12,13]. The main computing skill obtained by robotics-based learning is the sequence and conditionals, and we argue that the skill was augmented by logical thinking and mathematical operations supported by the math app.

## Section 6: Conclusion

We conducted a randomized controlled trial (RCT) to examine the impact of the interventions using educational robotics as well as math apps on primary students' performance in terms of computational thinking and mathematics, as well as general intelligence in Iraq. We show that the positive immediate impact is found for girls and is pronounced for poor-performing girls before the intervention. Moreover, a synergistic effect is observed whereby the impact on computational thinking is augmented by introducing math apps, thereby improving computational thinking further—and these two interventions also enhance general intelligence. Furthermore, a positive, sustainable impact is also found for girls who received both computer science and math education, suggesting their complementarity.

Our results show that computer science education using educational robots is effective in enhancing computational thinking and relevant skills in primary schools. Despite the growing awareness of the importance of computer science education, empirical evidence has remained scarce, especially regarding young learners. Future studies should advance the literature on empirical evidence on computer science education in several ways. First, a longer effect is worth investigating to confirm that students absorb knowledge of computer science—a goal that fell beyond the scope of this study. Second, any combination of computer science with other education programs should be examined to strengthen the impact of the programming workshop, in addition to the evidence of this study on the synergistic effect of computer science and math apps. Third, gender differences in the impact of computer science education should be further examined. Investigations such as this one in many countries will be informative both for academics and policymakers to advance computer science education globally.

## Acknowledgments

This study was conducted by the JICA Ogata Sadako Research Institute for Peace and Development (JICA Ogata Research Institute). We thank the JICA Ogata Research Institute for allowing us to use the data for this study and the JICA Iraq Office for the cooperation in data collection. We appreciate the many valuable suggestions from María Zapata-Cáceres, Estefanía Martín-Barroso and Marcos Román-González. We also wish to thank Hana Ashida for her research assistance. Our thanks also go to Alyakoub Mohannad, who made significant efforts to implement the program in Iraq, and the JICA Human Development Department, which supported the development of the math app. The views expressed in the paper are those of the authors and do not represent the official positions of JICA. The authors are responsible for any errors or omissions.

## Author contributions

**Conceptualization:** Satoshi Shimizutani, Shimpei Taguchi, Hiroyuki Yamada.

**Data curation:** Shimpei Taguchi.

**Formal analysis:** Shimpei Taguchi.

**Project administration:** Shimpei Taguchi.

**Validation:** Hiroyuki Yamada.

**Writing – original draft:** Satoshi Shimizutani.

**Writing – review & editing:** Shimpei Taguchi, Hiroyuki Yamada.

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
