## [Decision Letter · Decision Letter 0]

4 Apr 2025

The impact of computer science education in primary schools: Evidence from a randomized controlled trial in Iraq

PONE-D-24-56890

Dear Dr. Taguchi,

We’re pleased to inform you that your manuscript has been judged scientifically suitable for publication and will be formally accepted for publication once it meets all outstanding technical requirements.

Kind regards,

Henri Tilga, PhD

Academic Editor

PLOS ONE

Reviewers' comments:

Reviewer's Responses to Questions

**Comments to the Author**

1. Is the manuscript technically sound, and do the data support the conclusions?

Reviewer #1: Yes

2. Has the statistical analysis been performed appropriately and rigorously? 

Reviewer #1: Yes

3. Have the authors made all data underlying the findings in their manuscript fully available?

Reviewer #1: No

4. Is the manuscript presented in an intelligible fashion and written in standard English?

Reviewer #1: Yes

5. Review Comments to the Author

Reviewer #1: The study presents the results of a trial aimed at determining the impact on primary school students’ performance of using educational robotics tools and computer-assisted mathematics exercises.

The authors report several noteworthy empirical insights. Firstly, the short-term effect of robotics-based learning on the development of computational thinking is both positive and statistically significant among female students—particularly those identified as low-performing—while no significant effect is observed among male students. Secondly, the enhancement of computational thinking is further amplified through the integration of a mathematics application, which serves to reinforce computational competencies. Collectively, these two interventions also contribute to an improvement in general intelligence. Thirdly, the positive outcomes remained evident more than three months following the interventions among the cohort of female students who received instruction in both computer science and mathematics, thereby indicating a complementary relationship between these educational approaches.

The results presented demonstrate that teaching computer science through educational robotics at the primary school level is effective in enhancing computational thinking and related skills.

I consider that the article is technically rigorous and meets the scientific and ethical standards for inclusion in the journal.

6. PLOS authors have the option to publish the peer review history of their article (what does this mean? ). If published, this will include your full peer review and any attached files.

**Do you want your identity to be public for this peer review?** For information about this choice, including consent withdrawal, please see our Privacy Policy .

Reviewer #1: No

---

## [Editor Report · Acceptance letter]

PONE-D-24-56890

PLOS ONE

Dear Dr. Taguchi,

I'm pleased to inform you that your manuscript has been deemed suitable for publication in PLOS ONE. Congratulations! Your manuscript is now being handed over to our production team.

Kind regards,

on behalf of

Dr. Henri Tilga

Academic Editor

PLOS ONE